# A Progressive Feature Learning Network for *Cordyceps sinensis* Image Recognition

**DOI:** 10.3390/s25227082

**Published:** 2025-11-20

**Authors:** Shangdong Liu, Wenxiang Wu, Haijun Chen, Shuai You, Jiahuan Lu, Lin Mao, Fan Zhang, Yimu Ji

**Affiliations:** 1School of Computer Science, Nanjing University of Posts and Telecommunications, Nanjing 210023, China; lsd@njupt.edu.cn (S.L.); 1223045119@njupt.edu.cn (W.W.); 2Newlixon TechGroup.Co., Ltd., Nanjing 210012, China; chenhaijun@newlixon.com (H.C.); maolin@newlixon.com (L.M.); 3School of Internet of Things, Nanjing University of Posts and Telecommunications, Nanjing 210023, China; youshuai_666@163.com (S.Y.); 2023070801@njupt.edu.cn (J.L.); 4State Key Laboratory of Ocean Sensing, Ocean College, Zhejiang University, Zhoushan 316021, China; f.zhang@zju.edu.cn

**Keywords:** *Cordyceps sinensis*, feature mining, progressive learning, image recognition

## Abstract

*Cordyceps sinensis* (*C. sinensis*) is a valuable herbal medicine with wide-ranging applications. However, automating *C. sinensis* recognition is challenging due to the high morphological similarity and limited phenotypic variation among its subspecies. In this paper, we propose a novel approach called Progressive Feature Learning Network (PFL-Net) that mines multiple biological features to recognize different subspecies. Firstly, to comprehensively capture multi-scale discriminative features of *C. sinensis*, we propose the Spatial-aware Semantic Refinement Module (SSRM), which constructs discriminative feature groups by utilizing relative positions to model the intrinsic feature relations. Secondly, the Multi-scale Collaborative Perception Module (MCPM) avoids isolated biological features during modeling by establishing relations between different feature groups to enhance the recognition integrity of *C. sinensis*. Furthermore, to prevent the model from focusing on the same discriminative regions of *C. sinensis*, we propose a Channel Decouple (CD) loss that decouples features along the channel dimension, enhancing the diversity of *C. sinensis* discriminative features. In addition, we construct a *C. sinensis* dataset (CSD) to facilitate the application of biometric recognition, representing the first study focused on fine-grained *C. sinensis* recognition. Extensive experiments conducted on the CSD and three benchmark datasets validate the effectiveness of our proposed method, achieving a top-1 accuracy of 94.43% on the CSD dataset, which surpasses all existing approaches.

## 1. Introduction

*Cordyceps sinensis* (*C. sinensis*) is a rare and expensive medicinal herb, abundant in bioactive substances such as nucleotides, amino acids, polysaccharides, flavonoids, and sterols. Cordycepin, the primary constituent of authentic *C. sinensis*, is an alkaloid with antioxidant properties and exhibits pharmacological effects, including immune enhancement, antifatigue, and antitumor activities [1,2,3]. In Figure 1a, we present authentic and counterfeit *C. sinensis* samples and six core discriminative regions. Recognition of authentic *C. sinensis* from counterfeits poses a significant challenge, resulting in a proliferation of counterfeit products that pose a potential risk to public health. Existing recognition methods fall into two main categories. The first relies on expert experience, while the second employs multi-spectral and chemical analyses, which are both time-consuming and costly. With the advancement of computer vision [4,5,6], automatic image classification provides a new way to recognize *C. sinensis* accurately. To the best of our knowledge, this is the first study on image-based recognition of *C. sinensis*, which holds great significance for biomedicine.

*C. sinensis* possesses six primary discriminative features: the head, eyes, dorsal loops, front legs, middle legs, and tail legs. The recognition of *C. sinensis* based on expert experience requires consideration of all six biological features. Constructing multi-scale feature groups to establish discriminative regions is essential. As shown in Figure 1b, where each feature group represents a distinct discriminative region. After establishing multi-scale feature groups following expert experience, these feature groups need to be related. In Figure 1c, we establish relations between feature groups to ensure the integration of discriminative regions. This manner fully utilizes multiple regions of *C. sinensis* and enhances feature diversity. Different discriminative feature groups may focus on the same region, which could lead to redundancy in feature learning and hinder the ability to capture diverse and unique characteristics completely. As shown in Figure 1d, $G_{1}$ and $G_{1}^{\prime }$ both represent the “Eyes–Head–Dorsal loops–Front legs” features, which means they belong to the same discriminative region. Therefore, it is necessary to guide the network to focus on the six core biometric features of *C. sinensis*.

To address this issue, we propose a Progressive Feature Learning Network (PFL-Net), including the Spatial-aware Semantic Refinement Module (SSRM), the Multi-scale Collaborative Perception Module (MCPM), and the Channel Decouple (CD) loss. The SSRM constructs multi-scale feature groups $G_{1}$ to $G_{n}$. $G_{1}$ represents the relation between local features such as “Eyes–Head–Dorsal loops–Front legs”. $G_{n}$ represents the relation between larger scale features such as “Eyes–Head–Dorsal loops–Front legs–Middle legs–Tail legs”. The extraction of discriminative features is enhanced by modeling the relations among the six biometric features. The MCPM is designed to establish effective relations between different feature groups. This manner prevents information loss and ensures the integrity of discriminative features. The CD operates on each feature group to further decouple features within the group, alleviating the feature coupling problem in the modeling process and guiding the network to focus on the six core biometric features of *C. sinensis*.

Notably, to facilitate the recognition of *C. sinensis*, we constructed the first *C. sinensis* dataset (CSD) with 17k images, significantly surpassing the scale of existing fine-grained public datasets. CSD includes 27 categories of *C. sinensis*, covering samples from six different regions and counterfeit. In addition, we collect images from multiple perspectives to overcome the limitations of single-angle views in capturing the *C. sinensis* features. In summary, the main contributions of our work are:(1)We propose PFL-Net, capable of accurately extracting and relating discriminative features. To the best of our knowledge, PFL-Net is the first study on recognizing *C. sinensis*.(2)The SSRM is designed to model spatial contextual, mining multi-scale discriminative features of *C. sinensis*. The MCPM relates the feature extracted at multiple scales to avoid the loss of *C. sinensis* features.(3)The CD loss decouples the features within the channel dimension and guides the network to focus on the *C. sinensis* features.(4)*C. sinensis* has significant medicinal value, we construct the first dataset for CSD. We perform extensive experiments on CSD and three fine-grained classification benchmarks demonstrating the superior performance of PFL-Net.

## 2. Related Work

### 2.1. General Image Classification

In recent years, CNN demonstrated significant potential in image classification tasks, giving rise to several representative CNN architectures, including Res2Net [7], EfficientNetV2 [8], RepVGG [9], ConvNext [10], and ConvNextV2 [11]. These models focused on classification accuracy and network scalability. Res2Net [7] improved optimization by using identity mappings for signal propagation and multi-scale convolutions to expand the receptive field. EfficientNetV2 [8] introduced a progressive learning method that improved training speed and accuracy. RepVGG [9] used reparameterization techniques to balance classification precision and performance. ConvNext [10] added independent downsampling layers to improve model stability in image classification. ConvNextV2 [11] optimized self-supervised learning for image classification by integrating neural architecture design with masked autoencoders. These methods performed well in image classification tasks but are unsuitable for fine-grained problems with large intra-class variance and small inter-class differences.

### 2.2. Fine-Grained Image Classification

Fine-grained image recognition aims to recognize subtle differences between objects within a supercategory, such as different subspecies of animals or cars. Existing methods are divided into three categories: bounding box annotation-based, local-based [12,13,14], and attention-based methods. First, bounding box annotation-based methods were fully supervised and required bounding box annotations in both the training and testing stages. They focused on learning discriminative mid-level features and performed well in bird species recognition and face verification. Second, local-based methods focused on learning local embeddings by predicting masked or erased regions in the image. For example, MaskCOV [12] used a covariance matrix to capture the mutual information between each quarter of randomly masked and shuffled image patches. Third, motivated by the success of attention in visual classification, TransFG [15] enhanced global discrimination by paying attention to important markers, alleviating the challenge of category variation. Notably, existing methods demonstrate strong performance on public datasets, they are suboptimal for recognizing *C. sinensis*, which exhibits complex biological characteristics and structurally similar variants.

## 3. Method

### 3.1. Overview Architecture

In Figure 2a, the proposed Progressive Featuer Learning Network (PFL-Net) is composed of a backbone ResNet50 [4], SSRM, and MCPM. Let *F* represent our backbone feature extractor, which consists of *N* stages. The output feature map from any intermediate stage is denoted as $F^{n}$, with n={1,2,…,N}.

The SSRM applies at each intermediate stage of the backbone to extract feature maps. We use the output $F^{n}$ of the *n*-th intermediate stage of the backbone as the input of the SSRM module, denoted as $V^{n}=\mathrm{SSRM}\left(F^{n}\right)$. The output vector $V^{n}$ of each stage then passes through MCPM, denoted as $\left\{X^{n},X^{n-1}\right\}=\mathrm{MCPM}\left(V^{n}\right)$.

The classification module $f_{cls}^{n}$, consisting of two fully connected layers, predicts the probability distribution over classes for the n-th stage, denoted as $y^{n}=f_{cls}^{n}\left(X^{n}\right)$. The features extracted at the shallow level cannot fully demonstrate the biological features of Cordyceps sinensis, and the mining ability is limited. Therefore, we consider the final 3 stages: n=n,n−1,n−2. Lastly, we concatenate the outputs from the last three stages as follows [4,16]:(1)$$y^{cat}=f_{cls}^{cat}\left(concat\left[X^{n-2},X^{n-1},X^{n}\right]\right)$$

### 3.2. Spatial-Aware Semantic Refinement Module

CNN extracts structural features through convolution and gradually integrates features across layers to form a multi-level representation. Spatial structural information is crucial for *C. sinensis* image recognition. As shown in Figure 2b, we propose a Spatial-aware Semantic Refinement Module (SSRM) that builds feature groups by learning the spatial context information of the object, thereby enhancing the feature representation capability of the backbone network.

For the input image *I*, SSRM directly applies to the feature map f(I) extracted by the backbone. First, convolution is performed on the feature map to obtain $h\left(l\right)\in R^{C\times N\times N}$, which represents the spatial information of different features. The spatial relation of various features can be obtained by modeling the structural information between different parts of *C. sinensis*.

This paper uses polar coordinates to measure spatial relations between different regions. The traditional Cartesian coordinate system effectively measures linear distances but is sensitive to rotation and scale changes when modeling spatial relationships within an object [17]. In contrast, the polar coordinate system (r,θ) represents spatial layouts using radial distance and angle relative to a central point, offering inherent rotation invariance and scale adaptability. In biological image recognition, where structures such as the head, body, and tail of *Cordyceps sinensis* exhibit axial or radial organization, polar coordinates provide a more robust means of capturing geometric and topological relationships, ensuring stable modeling of structural coherence and spatial dependencies [18]. Given a reference region $R_{o}=R_{x,y}$, indexed at (x,y) on the N×N plane, and a reference horizontal direction, the polar coordinates of the region $R_{i,j}$ can be written as $\left(r_{i,j},\theta_{i,j}\right)$:(2)$$r_{i,j}=\frac{1}{N}\sqrt{\frac{{\left(x-i\right)}^{2}+{\left(y-j\right)}^{2}}{2}}$$(3)$$\theta_{i,j}=\frac{\left(atan2(y-j,x-i)+\pi \right)}{2\pi }$$
where $r_{i,j}$ measures the relative distance. $\theta_{i,j}$ measures the polar angle of $R_{i,j}$ relative to the horizontal direction. atan2(·) is a function that calculates the angle between two points. In this paper, we select the region with the maximum response m(I) as the reference region:(4)$$R_{o}=R_{x,y},\;\left(x,y\right)=\arg\max_{1\leq x,y\leq N}m^{\prime }{\left(I\right)}_{i,j}$$

The SSRM utilizes polar coordinates to guide the module in learning and identifying the features of *C. sinensis*. Specifically, the SSRM module calculates the polar coordinates of the region $R_{i,j}$ by analyzing the discriminant information of the target region $R_{i,j}$ and the reference region $R_{0}$. This process involves channel-wise concatenation of the feature map h(l) with ${h(I)}_{x,y}$, followed by a fully connected layer to generate the predicted polar coordinates $\left(r_{i,j}^{\prime },\theta_{i,j}^{\prime }\right)$. The SSRM module models the spatial structure between different parts of an object and integrates the object region mask $m^{\prime }\left(I\right)$ learned from the backbone network.

The SSRM first measures the relative distance differences between all regions and the object:(5)$$L_{d}=\sum_{I\in I}\sqrt{\frac{\sum_{1\leq i,j\leq N}m^{\prime }{\left(I\right)}_{i,j}{\left(r_{i,j}^{\prime }-r_{i,j}\right)}^{2}}{\sum m^{\prime }\left(I\right)}}$$

The second measures the angular difference between regions within the object. The structural information of an object should be rotationally invariant and robust to various appearances and poses. Therefore, we calculate the angle loss $L_{a}$ using the standard deviation of the difference between the predicted and true polar angles, as follows:(6)$$L_{a}=\sum_{I\in I}\sqrt{\frac{\sum_{1\leq i,j\leq N}m^{\prime }{\left(I\right)}_{i,j}{\left(\theta_{\Delta_{i,j}}-{\bar{\theta }}_{\Delta }\right)}^{2}}{\sum m^{\prime }\left(I\right)}}$$(7)$$\theta_{\Delta_{i,j}}=\left\{\begin{array}{cc} \theta_{i,j}^{\prime }-\theta_{i,j}, & \mathrm{if}\;\theta_{i,j}^{\prime }-\theta_{i,j}\geq 0\\ 1+\theta_{i,j}^{\prime }-\theta_{i,j}, & \mathrm{otherwise}, \end{array}\right.$$(8)$${\bar{\theta }}_{\Delta }=\frac{1}{\sum m^{\prime }\left(I\right)}\sum_{1\leq i,j\leq N}m^{\prime }{\left(I\right)}_{i,j}\theta_{\Delta_{i,j}}$$
where ${\bar{\theta }}_{\Delta }$ represents the average difference between the predicted polar angle and the actual polar angle. SSRM models the relative structure between object parts. During polar coordinate regression, the predicted semantic mask $m^{\prime }\left(I\right)$ filters out irrelevant visual information outside the main object. In general, the loss function of SSRM can be expressed as:(9)$$L_{ssrm}=L_{d}+L_{a}$$

The polar-coordinate-based losses supervise the network to learn structural consistency between predicted and ground-truth spatial relations. Specifically, the distance loss $L_{d}$ penalizes deviations in the relative radial distances between regions, ensuring scale consistency. The angular loss $L_{a}$ constrains the predicted angular relations to maintain rotational invariance and correct part ordering. Jointly minimizing $L_{d}+L_{a}$ enforces the backbone to encode spatial geometry of *C. sinensis* parts in a biologically meaningful manner, improving robustness to pose and orientation variations. Through SSRM, the backbone learns and recognizes the structural features of the object, enabling the backbone network to model the spatial dependencies between the parts of the object and mine features.

### 3.3. Multi-Scale Collaborative Perception Module

Given the importance of learning discriminative and diverse features in Fine-grained Image Classification (FGIC) [18,19,20,21], we propose the Multi-scale Collaborative Perception Module (MCPM). The module enhances the expressiveness of local features by aggregating complementary information from different scales, improving feature discriminability and diversity.

Figure 2c briefly illustrates the structure of the MCPM. To demonstrate the Effectiveness of the approach, we denote two different scale feature representations as $X_{s_{1}}\in R^{C\times W_{1}H_{1}}$ and $X_{s_{2}}\in R^{C\times W_{2}H_{2}}$. The subscript $s_{i}$ indicates that $X_{s_{i}}$ focuses on the *i*th part of the object. We regard the feature vector at each spatial location in the channel dimension as a pixel, as follows:(10)$$px\left(X,i\right)={\left(X_{1,i},\ldots ,X_{C,i}\right)}^{T}$$
where px stands for pixels. We first compute the similarity between the pixels in $X_{s_{1}}$ and those in $X_{s_{2}}$:(11)$$M=f\left(X_{p_{1}},X_{p_{2}}\right),\;f\left(X,Y\right)=X^{T}Y$$

We use the inner product to compute the similarity. The $M_{i,j}$ represents the similarity between the *i*th pixel of $X_{s_{1}}$ and the *i*th pixel of $X_{s_{2}}$. The lower the similarity between two pixels, the greater their complementarity. Therefore, we use *M* as the complementarity matrix. We then normalize *M* along both the row and column directions, as follows:(12)$$\begin{array}{cc} A_{s_{1}}^{s_{2}} & =-œ\left(M^{T}\right)\in \left[0,1\right]\\ A_{s_{2}}^{s_{1}} & =-œ(M)\in [0,1] \end{array}$$
where œ is softmax, the operation is applied column-wise. This allows us to obtain the complementary information:(13)$$\begin{array}{c} Y_{s_{1}}^{s_{2}}=X_{s_{2}}A_{s_{1}}^{s_{2}}\in R^{C\times W_{1}H_{1}}\\ Y_{s_{2}}^{s_{1}}=X_{s_{1}}A_{s_{2}}^{s_{1}}\in R^{C\times W_{2}H_{2}} \end{array}$$
where $Y_{s_{j}}^{s_{i}}$ represents the complementary information of $X_{s_{i}}$ with respect to $X_{s_{j}}$. Each pixel of $Y_{s_{2}}^{s_{1}}$ can be written as:(14)$$px\left(Y_{s_{1}}^{s_{2}},i\right)=\sum_{j\in [1,W_{2}H_{2}]}{\left(A_{s_{1}}^{s_{2}}\right)}_{i,j}\times px\left(X_{p_{2}},j\right)$$

Each pixel in $Y_{s_{1}}^{s_{2}}$ uses all pixels of $X_{s_{2}}$ as a reference. The higher the complementarity between $px(X_{s_{1}},i)$ and $px(X_{s_{2}},j)$, the greater the contribution of $px(X_{s_{2}},j)$ to $px(Y_{s_{1}}^{s_{2}},i)$. Thus, each pixel within these scale features can capture semantically complementary information from other pixels. In the normal case, given a set of part-specific features $S=\{X_{s_{1}},X_{s_{2}},X_{s_{3}},\ldots ,X_{s_{n}}\}$, the complementary information of $X_{s_{i}}$ is:(15)$$Y_{s_{i}}=\sum_{X_{s_{j}}\in P\wedge i\neq j}Y_{s_{i}}^{s_{j}}$$

The $Y_{s_{1}}^{s_{2}}$ can be obtained by applying $X_{s_{i}}$ and $X_{s_{j}}$ to Equations (Equation 13), (Equation 14) and (Equation 16). We can compute both $Y_{s_{i}}^{s_{j}}$ and $Y_{s_{j}}^{s_{i}}$ simultaneously. This results in the enhanced object features:(16)$$Z_{s_{i}}=X_{s_{i}}+\gamma \times Y_{s_{i}}$$
where γ is a hyperparameter controlling the diversification degree, default set to 2.

### 3.4. Loss Function

During feature learning, excessive focus on the same discriminative regions can occur. To address this, we propose a Channel Decouple (CD) loss, which decouples features and mitigates feature coupling. After inputting an image into the network, we extract the feature map, denoted as $F\in R^{C\times W\times H}$, with height *H*, width *W*, and number of channels *C*. We need to set the value of *C* equal to c×k, where *c* and *k* indicate the number of classes in a dataset and the number of feature channels used to represent each class. The *n*th vector feature channel of *F* is represented as $F_{n}\in R^{WH},n=1,2,\ldots ,N$. Each channel matrix of *F* is reshaped into a vector of size WH. The grouped feature channels corresponding to the *i*th class are represented as $F_{i}\in R^{k\times WH},i=0,1,\ldots ,c-1$. This can be expressed as:(17)$$F_{i}=F_{i\times k+1},F_{i\times k+2},...,F_{i\times k+k}$$

The feature group $F=\{F_{0},F_{1},\ldots ,F_{c-1}\}$ is processed through two parallel streams in the network, each designed with a distinct sub-loss tailored for two different objectives. In the cross-entropy stream, *F* is treated as the input to a fully connected layer with the traditional CE [22]. The CE encourages the network to extract informative features focused primarily on global discriminative regions. On the other hand, the CD stream supervises the network to highlight different local discriminative regions. A specific number of grouped feature channels is representative of each class. The discriminative component enforces class alignment among the feature channels, ensuring that each feature channel corresponding to a specific class has sufficient discriminative power. $L_{CD}$ can be expressed as: (18)$$L_{CD}\left(F\right)=L_{CE}\left(y,\frac{1}{\sum_{i=0}^{c-1}e^{g(F_{i})}}{\left[e^{g(F_{0})},\ldots ,e^{g(F_{c-1})}\right]}^{T}\right)$$
where g(·) is defined as:(19)$$g\left(F_{i}\right)=\frac{1}{WH}\sum_{k=1}^{WH}\max_{j=1,2,\ldots ,\xi }\left[M_{i}\cdot F_{i,j,k}\right]$$
where $M_{i}$ is a random mask between 0 and 1. The cross-entropy (CE) and Channel Decouple (CD) losses are optimized jointly in a parallel manner. The backbone features are simultaneously fed into two loss branches: one for global discrimination (CE stream) and one for local feature diversification (CD stream). During training, the gradients from both loss functions are backpropagated to the shared feature extractor, with the total loss formulated as(20)$$Loss\left(F\right)=L_{CE}\left(F\right)+\mu \times L_{CD}\left(F\right)$$

The weighting coefficient μ controls the relative influence of the CD loss. Empirically, we set μ = 0.3 to balance global classification accuracy and feature diversity. This joint optimization ensures that the model simultaneously enhances discriminability and suppresses channel redundancy.

## 4. Data Collection and Construction

### 4.1. Material Preparation

We purchased and collected 654 *Cordyceps sinensis* (*C. sinensis*) samples, covering six major production regions and including counterfeit samples. The pharmacological efficacy of *C. sinensis* is strongly influenced by its place of origin, with the six major production regions being Yushu, Guoluo, Haixi, Hainan, Haibei, and Huangnan. As shown in Figure 3, we present two sample images for each specification of *C. sinensis* from each production area. Counterfeit samples refer to artificially synthesized *C. sinensis*, which have no medicinal value and may harm health. Each region’s *C. sinensis* samples vary in size, defined by the number of specimens per 500 g. Smaller sizes correspond to larger individual specimens, which generally have higher medicinal value. We then proceeded to capture images of these samples.

### 4.2. Data Collection and Annotation

Images are collected using eight smart devices, including both Android and iOS devices, covering a wide range of popular devices. The images of *C. sinensis* were taken from four angles: the back, the foot, the left side, and the right side. As shown in Figure 3 the last row. To ensure data consistency and minimize background interference, all images were captured using the camera’s nine-grid layout, with the *C. sinensis* positioned within the central three grids and the eyes aligned along the first horizontal line. A ring light was placed above the specimens to optimize imaging conditions and capture fine details, while a standardized black background was used to reduce background noise. The final dataset comprises over 17k images. Table 1 presents the number of *C. sinensis* images collected for each specification from different production regions.

To ensure reproducibility, the CSD dataset was randomly divided into five balanced subsets using a fixed random seed (seed = 1024). Each subset preserves the class distribution across all 27 categories. During 5-fold cross-validation, four subsets were used for training and one for testing in each iteration, ensuring no overlap between training and test samples. This partitioning principle guarantees consistency across repeated experiments.

#### Environmental Considerations and Optical Stability

Although the dataset was collected under controlled illumination using a ring light and standardized background, the quality of captured images may still be influenced by subtle environmental factors such as air turbulence, humidity, and temperature fluctuations during shooting. These conditions can induce wavefront aberrations and minor defocus effects that alter the sharpness and spatial coherence of fine-grained texture details. Previous optical studies [23] have demonstrated that turbulent environments can distort the wavefront propagation of light, leading to random phase errors and reduced contrast in microscopic structures. While our imaging setup minimizes such disturbances, potential optical aberrations may still affect the precision of spatial feature extraction, especially for thin or reflective specimens of *C. sinensis*. In future work, incorporating adaptive optics or turbulence-aware image correction techniques could further improve the robustness of fine-grained feature acquisition under non-ideal environmental conditions.

## 5. Experiments

### 5.1. Datasets and Settings

**Data.** In the CSD, we use 70% of the images for training and the remaining 30% for testing, consistent with the public datasets. To verify the generalization ability of our proposed PFL-Net, we conducted comprehensive experiments on three public fine-grained recognition benchmarks: CUB-200-2011 (CUB) [24], Stanford Cars (CAR) [25], and FGVC-Aircraft (AIR) [26]. In Table 2, we followed the standard procedure for splitting the training and testing images and used top-1 accuracy as the evaluation metric.

**Implementation Details.** We used ResNet50 [4] as backbone networks. In the experiment, we only used image labels for supervised training, random cropping, horizontal flipping augmentation during training, and center cropping during testing. The input image size is 448 × 448. The SGD optimizer is used with a momentum of 0.9 and weight decay 5 × 10^−4^. The learning rate of the backbone layer is 0.0002, and the learning rate of the new layer is 0.002, both adjusted according to the cosine annealing strategy [27]. The learning rate of the auxiliary classifier is kept constant at 0.01. The model is trained for 200 rounds and is trained end-to-end on Nvidia RTX 3090 (Nvidia, Santa Clara, CA, USA) based on PyTorch 2.1.1.

### 5.2. Comparison with State of the Arts

Table 3 presents the experimental results of the comparative evaluation on the CSD, CUB-200-2011, Stanford Cars, and FGVC-Aircraft. We proposed that PFL-Net consistently outperforms state-of-the-art methods on both the CSD dataset and the three widely used benchmarks. The enhancement in performance is due to the design of our network and the efficiency of its components.

PFL-Net surpasses methods that leverage implicit data augmentation, such as DCL [38], ISDA [28], and LearnableISDA [31], demonstrating its superior ability to capture fine-grained details. In addition, PFL-Net outperforms attention-based methods like S3Ns [36], ACNet [34], AP-CNN [33], and P2P-Net [40]. While PMG [37] achieved promising results by aggregating images, PFL-Net further enhances the learning of inter-class differences by mining multi-scale features and fusing information across scales via the MCPM. PFL-Net significantly outperforms PMG [37] in classification accuracy. Compared to API-Net [35], PFL-Net achieves superior classification performance without the need to construct specific image pairs. Furthermore, PFL-Net outperforms Bi-FRN [30] and C2-Net [29] by eliminating the need to construct support-query sample pairs. It places greater emphasis on the relationships between features, resulting in generally superior classification performance.

### 5.3. Ablation Studies

To evaluate the effectiveness of the key components in our proposed approach, we conducted ablation experiments on CSD and CUB [24]. The results are shown in Table 4. The introduction of SSRM improves accuracy by +1.16% on CSD, demonstrating the benefit of spatial context modeling. MCPM further enhances accuracy by +1.39% through cross-scale feature collaboration. Finally, incorporating the CD loss provides an additional +1.39% gain by encouraging channel diversity and reducing redundant focus. Overall, PFL-Net achieves a total improvement of +2.79% over the baseline, validating the effectiveness of each component.

Specifically, as shown in Figure 4, SSRM provides a spatial context modeling manner, which helps to mine the discriminative features of *C. sinensis*. In addition, MCPM can associate the discriminative regions of *C. sinensis* to avoid information loss. The overall performance can be further improved owing to the complementary nature of SSRM and MCPM. Furthermore, we use CD loss to decouple features and guide the network to focus on the core discriminative features of *C. sinensis*. The results show that our Progressive Feature Learning Network can fully identify each feature region of *C. sinensis*.

### 5.4. Visualizations

**Class Activation Map**—We further apply Grad-CAM [41] to the final convolutional layer to provide intuitive visualizations. Figure 5 displays the activation maps for four datasets. Compared to the baseline model, PFL-Net focuses more on discriminative regions of the target object, such as the body of an airplane, the main contours of a car, the head of a bird, and the morphological features of *C. sinensis*. In addition, PFL-Net exhibits significantly lower activation in background areas, demonstrating its effectiveness in suppressing noise and irrelevant information. Visualizations across multiple datasets reveal that PFL-Net effectively extracts more category-specific regions, creating clearer boundaries between the object and background.

**Parts Location**—The discriminative regions identified by our PFL-Net are shown in Figure 6. Columns 1 to 3 display bounding boxes for only the first two parts, while columns 4 and 5 show all four primary discriminative regions. As illustrated, the first two regions exhibit notable similarities: for aircraft, they are the wings and tail; for birds, the head and body; for cars, the front and roof; for *C. sinensis*, they are the head and eyes. PFL-Net is capable of accurately and efficiently recognizing these key discriminative regions, demonstrating high robustness.

**Feature Visualization**—To intuitively evaluate the separability of features learned by the proposed PFL-Net, we employ the t-distributed Stochastic Neighbor Embedding (t-SNE) [42,43,44] to project high-dimensional features into a two-dimensional space for visualization, as shown in Figure 7. t-SNE is a nonlinear dimensionality reduction method that preserves the local neighborhood structure of data by modeling pairwise similarities between samples in both high-dimensional and low-dimensional spaces. Given two samples $x_{i}$ and $x_{j}$ in the high-dimensional feature space, their similarity is modeled by a conditional probability:(21)$$p_{j|i}=\frac{\exp\left(-∥x_{i}-x_{j}{∥}^{2}/2\sigma_{i}^{2}\right)}{\sum_{k\neq i}\exp\left(-∥x_{i}-x_{k}{∥}^{2}/2\sigma_{i}^{2}\right)},$$
where $\sigma_{i}$ controls the variance of the Gaussian kernel centered at $x_{i}$. In the low-dimensional embedding, the similarity between points $y_{i}$ and $y_{j}$ is defined as:


(22)
$$q_{ij}=\frac{{\left(1+∥y_{i}-y_{j}{∥}^{2}\right)}^{-1}}{\sum_{k\neq l}{\left(1+∥y_{k}-y_{l}{∥}^{2}\right)}^{-1}}.$$


t-SNE minimizes the Kullback–Leibler (KL) divergence between these two distributions:
(23)$$L_{t-\mathrm{SNE}}=\sum_{i}\sum_{j}p_{ij}\log\frac{p_{ij}}{q_{ij}},$$
ensuring that nearby points in the high-dimensional space remain close in the low-dimensional visualization, while distant points are pushed apart. This results in an embedding that effectively preserves local similarities and reveals cluster structures. As illustrated in Figure 7, the features extracted by PFL-Net form more compact intra-class clusters and exhibit clearer inter-class boundaries compared to the baseline, indicating superior discriminative feature representation and robustness in fine-grained classification tasks.

**Confusion Matrix**—As shown in Figure 8, the left matrix illustrates the confusion matrix of the baseline model, where the diagonal elements are scattered and many off-diagonal entries exhibit noticeable values, indicating frequent misclassifications among visually similar subspecies. This suggests that the baseline model struggles to learn discriminative representations for categories with subtle morphological differences. In contrast, the right panel presents the confusion matrix of PFL-Net, where nearly all the values are concentrated along the main diagonal, and off-diagonal elements are close to zero. This demonstrates that PFL-Net can accurately distinguish among all 27 *C. sinensis* categories, including those with highly similar appearances. The improved performance results from the integration of the Spatial-aware Semantic Refinement Module (SSRM) and Multi-scale Collaborative Perception Module (MCPM), which enable the model to better capture fine-grained spatial relationships and maintain feature consistency across scales. Overall, PFL-Net achieves clearer classification boundaries, lower confusion rates, and stronger generalization across subspecies.

## 6. Conclusions

This paper is the first to study the recognition task of Cordyceps sinensis, constructing a dedicated dataset and proposing a Progressive Feature Learning Network (PFL-Net) to promote the development of recognition in the biomedical field. By effectively combining the Spatial-aware Semantic Refinement Module (SSRM) and Multi-scale Collaborative Perception Module (MCPM), PFL-Net enhances the relation between local and global features, improving its ability to recognize complex structures. The Channel Decoupling (CD) loss further optimizes the network to extract diverse features. Experiments on Cordyceps sinensis dataset and three fine-grained classification benchmarks indicate that PFL-Net achieves state-of-the-art performance, and visualization results demonstrate its effectiveness. In the future, PFL-Net has the potential to be applied to other biological recognition scenarios.

### Limitations and Future Work

Although PFL-Net achieves promising results, several limitations remain. First, the model relies on high-quality annotations and balanced category distributions; significant noise or imbalance in the dataset may affect its stability. Second, while the Spatial-aware Semantic Refinement Module (SSRM) provides spatial robustness, extreme variations in illumination or occlusion may still degrade feature consistency. Third, the multi-scale collaborative design slightly increases computational cost compared to lightweight architectures. Future work will focus on optimizing model efficiency and extending PFL-Net to other biological recognition domains with more diverse imaging conditions.

## Figures and Tables

**Figure 1 sensors-25-07082-f001:**
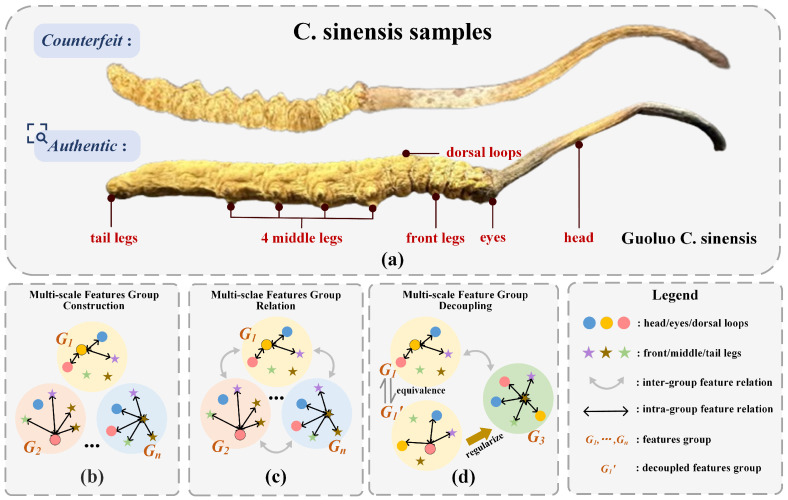
(**a**) illustrates authentic and counterfeit *C. sinensis* samples, highlighting six core discriminative features, while (**b**–**d**) detail the analysis process of multi-scale features of *C. sinensis*.

**Figure 2 sensors-25-07082-f002:**
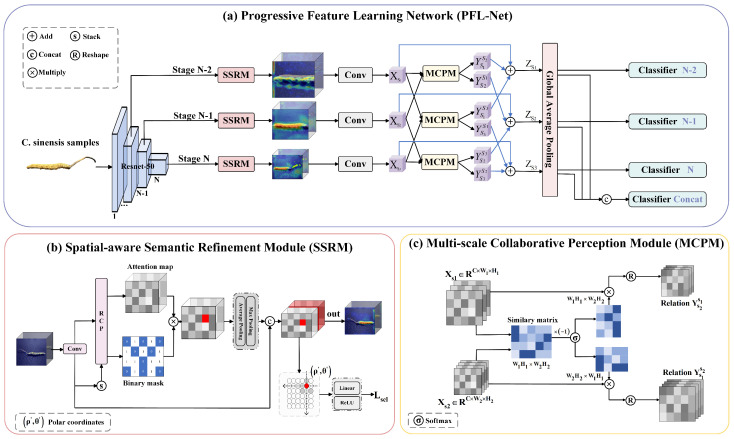
(**a**) The overall architecture of PFL-Net includes a feature extractor, the Spatial-aware Semantic Refinement Module(SSRM), and the Multi-scale Collaborative Perception Module(SSRM). “N” represents the number of stages in the backbone network. (**b**) shows the structure of the SSRM. (**c**) illustrates the MCPM.

**Figure 3 sensors-25-07082-f003:**
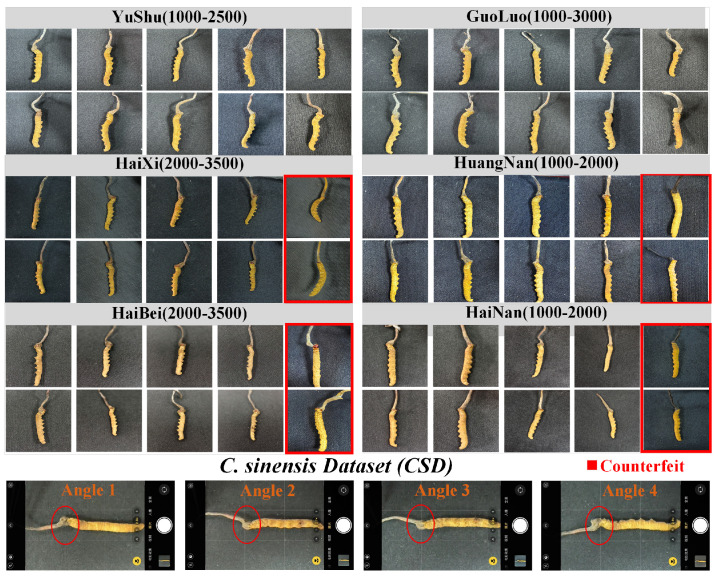
The *C. sinensis* dataset consists of *Cordyceps sinensis* samples from six major production regions, along with counterfeit samples. Dataset shooting rules are shown on the last line.

**Figure 4 sensors-25-07082-f004:**
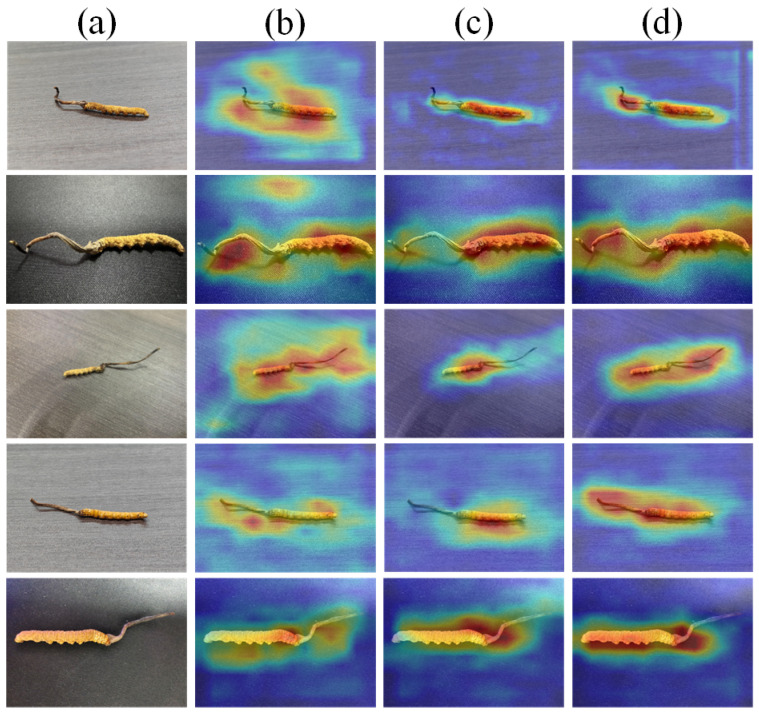
Visualization of ablation experiments. (**a**) input, (**b**) Baseline, (**c**) Baseline + SSRM + MCPM, and (**d**) PFL-Net.

**Figure 5 sensors-25-07082-f005:**
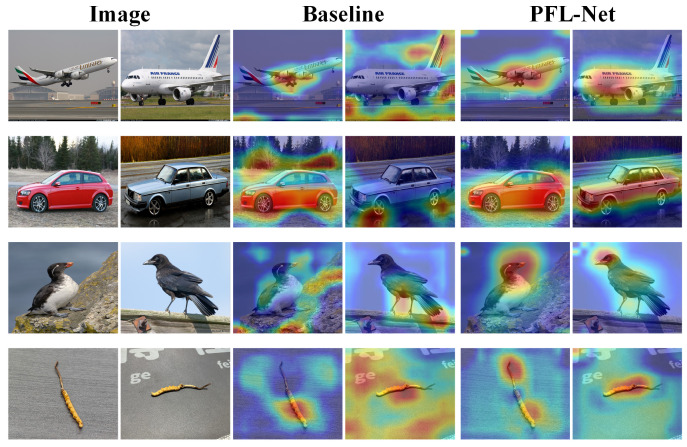
Class Activation Map on four datasets.

**Figure 6 sensors-25-07082-f006:**
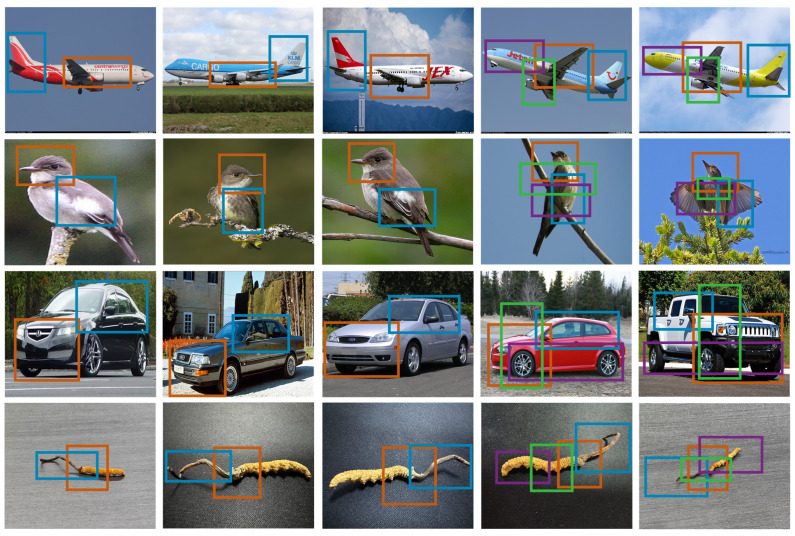
Discriminative regions detected by our PFL-Net.

**Figure 7 sensors-25-07082-f007:**
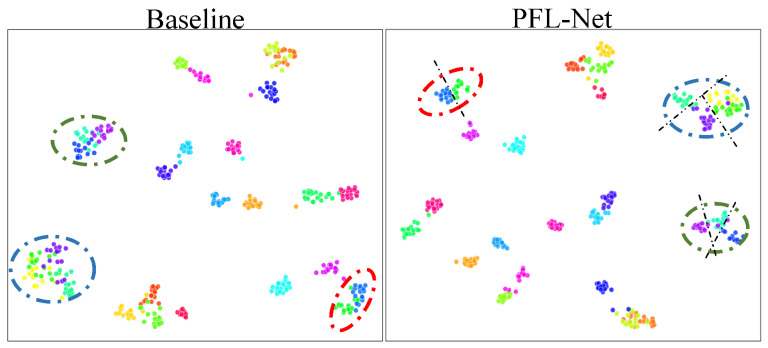
A t-SNE plot of learned representations on CSD.

**Figure 8 sensors-25-07082-f008:**
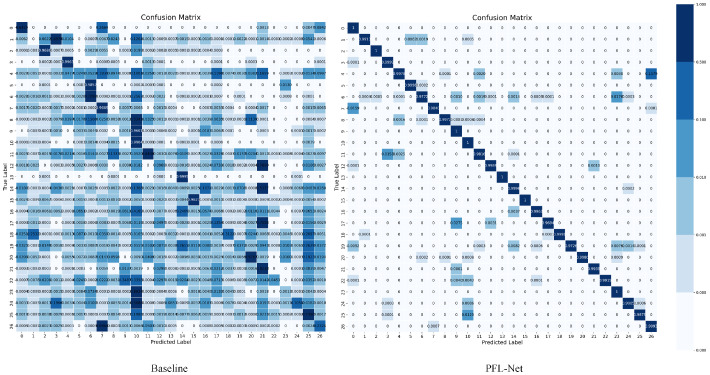
Confusion matrix on CSD.

**Table 1 sensors-25-07082-t001:** Distribution of *C. sinensis* products by origin, specification (number of pieces per 500 g), and number of images.

Origin	Specification	Image	Origin	Specification	Image
Yushu	1000	636	Guoluo	1000	600
	1200	626		1200	640
	1500	620		1500	640
	2000	568		2000	592
	2500	616		3000	576
Haixi	2000	636	Huangnan	1000	616
	2500	640		1200	584
	3000	616		1500	636
	3500	616		2000	640
Haibei	2000	616	Hainan	1000	640
	2500	640		1200	624
	3000	636		1500	600
	3500	640		2000	640
Counterfeit	/	3167			

**Table 2 sensors-25-07082-t002:** Overview of the public datasets used in our experiments, including the number of classes, training, and test samples for each dataset.

Dataset	Classes	Training	Testing
CUB-200-2011 [24]	200	5994	5794
Stanford Cars [25]	196	8144	8041
FGVC-Aircraft [26]	100	6667	3333

**Table 3 sensors-25-07082-t003:** The top-1 accuracy (%) comparison with state-of-the-art methods on the CSD, CUB-200-2011, Stanford Cars, and FGVC-Aircraft datasets is presented. Bold values indicate the best results, and underlined values represent the second-best results.

Method	Venue	*C. sinensis* Dataset	CUB-200-2011	Stanford Cars	FGVC-Aircraft
ISDA [28]	NeurIPS19	81.4	85.3	91.7	93.2
C2-Net [29]	AAAI24	88.5	84.6	-	88.9
Bi-FRN [30]	AAAI23	89.9	85.4	-	88.4
LearnableISDA [31]	TIP24	90.2	86.7	92.7	94.3
iSICE [32]	CVPR23	90.2	85.9	93.5	92.7
AP-CNN [33]	TIP21	90.5	87.2	92.2	93.6
ACNet [34]	CVPR20	90.6	88.1	92.4	94.6
API-Net [35]	AAAI20	91.0	87.7	93.0	94.8
S3Ns [36]	ICCV19	91.2	88.5	92.8	94.7
PMG [37]	ECCV20	92.1	88.9	92.8	95.0
DCL [38]	CVPR19	92.8	87.8	93.0	94.5
ViT [39]	ICLR21	93.1	90.3	94.2	94.8
P2P-Net [40]	CVPR22	93.2	90.2	94.9	94.2
TransFG [15]	AAAI22	93.7	**91.7**	94.8	-
**PFL-Net (our)**	-	**94.4**	91.2	**94.9**	**95.1**

**Table 4 sensors-25-07082-t004:** Ablation studies on the CSD and CUB (using original images) are conducted with the baseline (ResNet50) to evaluate the impact of different modules.

Index	Component				Accuracy (%)	
	Baseline	SSRM	MCPM	CD	CSD	CUB
0	√				91.64	88.88
1	√	√			92.80	89.02
2	√		√		91.87	89.93
3	√			√	93.03	89.41
4	√	√	√		93.34	90.95
5	√	√		√	94.19	90.55
6	√		√	√	93.75	90.41
7	√	√	√	√	**94.43**	**91.26**

## Data Availability

The data presented in this study are available on request from the corresponding author.

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
