# Peer review of "A Progressive Feature Learning Network for Cordyceps sinensis Image Recognition"

_sensors, 2025, doi:10.3390/s25227082_

Round 1

Reviewer 1 Report

Comments and Suggestions for Authors

This manuscript propose a novel approach called Progressive Feature Learning Network (PFL- 5 Net) to recognize different subspecies. This topic has significant practical value, but there are still multiple issues to be addressed in this manuscript. Specific comments are as follows:
1,This paper uses polar coordinates to measure spatial relations between different regions,it is necessary to elaborate on the necessity and value of using polar coordinates.
2,Abbreviations need only to be defined once in the text, and can be used directly thereafter, such as e C. sinensis Dataset (CSD) .....
3,The text in many images in the manuscript is difficult to read, such as Figure 8
4,Figure 7 presents a t-SNE scatter plot. Here, it is necessary to explain the content and mathematical principle of the t-SNE method

Reviewer 2 Report

Comments and Suggestions for Authors

The paper presents a comprehensive study on the recognition of Cordyceps sinensis using PFL-Net. The method demonstrates high effectiveness but requires some clarifications. The following points need clarification before publication.

Questions and Remarks:

  1. The architectural integration and joint optimization of the cross-entropy and CD loss streams need to be clarified.
  2. The principle for partitioning the CSD dataset into subsets should be described to ensure reproducibility.
  3. Quantitative results of the ablation studies are recommended to evaluate the contribution of SSRM, MCPM, and CD.
  4. The mechanism for calculating losses based on polar coordinates in the SSRM module should be detailed.
  5. The universality of PFL-Net for other biological recognition tasks should be substantiated by specifying which types of features the method is oriented towards.

Concluding Remarks:

  1. It is recommended to provide a brief discussion on the potential limitations of the current study or the proposed method.
  2. It is necessary to describe the potential influence of wavefront aberrations, including turbulent environment, on the quality of image recognition (based on an article like https://doi.org/10.1016/j.optlastec.2025.113342).
  3. The authors are advised to thoroughly proofread the manuscript for minor grammatical errors to enhance readability.

Minor revision.

Reviewer 3 Report

Comments and Suggestions for Authors

Title of scientific manuscript:

"A Progressive Feature Learning Network for Cordyceps sinensis Image Recognition"

Introduction:

The attached scientific study made by a group of authors in an innovative way wants to recognize the medicinal plant Cordyceps sinensis suggesting and using the Progressive Feature Learning Network CNN (PFL-Net). The authors emphasize that this is the first scientific study for the plant in question.

A scientific manuscript is properly structured and written, supported by figures and tables, both qualitatively and quantitatively, and therefore has an appropriate scientific structure.

The contributions of scientific work using designed methods and procedures were demonstrated. Reference list is adequate. Authors presents a t-SNE scatter plot (Figure 7) based on the learned high-dimensional features compared to the baseline for the purpose of proving that PFL-Net shows significantly better feature representation.

Overview:

The introduction is complete and provides an adequate overview of the topic, the section on previous works and the description are relevant and referential, the method is described fundamentally and correctly, the data collection system is correct, and the experimental results are presented in numerous tables and figures.

Remarks:

A chapter on motivation and methodology is missing.

The reference for equation (1) is missing.

No other remarks.

Conclusion:

The group of authors in this innovative approach encourages the development of recognition methods based on CNN platforms in the biomedical field. In this case, plants of significant medicinal value can be easily detected using the presented methods, avoiding traditional multi-spectral and chemical analyses, which are both time-consuming and expensive.

Reviewer 4 Report

Comments and Suggestions for Authors

The study proposes a deep learning model, (PFL-Net), to automatically recognize subspecies of Cordyceps sinensis, a medicinal fungus with highly similar morphology. The model integrates multiple modules to capture multi-scale biological features, and presents a new C. sinensis dataset, achieving good recognition performance across several benchmarks. There are several points that should be addressed to improve the manuscript quality:

  • The abstract Lacks a clear logical flow (problem → method → results → impact).
  • Method details dominate the abstract, while motivation and results are underdeveloped.
  • I general, there is a lack of of evaluation metrics, baseline comparisons, or percentage improvements.
  • The authors fails to highlight how this work advances the state of the art compared to prior deep learning models.
  • Some sentences are long and complex, making the text difficult to follow.
